# Ethnoveterinary Practices and Ethnobotanical Knowledge on Plants Used against Cattle Diseases among Two Communities in South Africa

**DOI:** 10.3390/plants11131784

**Published:** 2022-07-05

**Authors:** Mompati Vincent Chakale, John Awungnjia Asong, Madeleen Struwig, Mulunda Mwanza, Adeyemi Oladapo Aremu

**Affiliations:** 1Indigenous Knowledge Systems Centre, Faculty of Natural and Agricultural Sciences, North-West University, Private Bag X2046, Mmabatho 2790, South Africa; 25279076@g.nwu.ac.za; 2School of Philosophy, Faculty of Humanities, North-West University, Private Bag X2046, Mmabatho 2790, South Africa; 3Unit for Environmental Sciences and Management, Faculty of Natural and Agricultural Sciences, North-West University, Private Bag X1290, Potchefstroom 2520, South Africa; johnmilan058@gmail.com (J.A.A.); madeleen.struwig@nwu.ac.za (M.S.); 4Food Security and Safety Niche Area, Faculty of Natural and Agricultural Sciences, North-West University, Private Bag X2046, Mmabatho 2790, South Africa; mulunda.mwanza@nwu.ac.za; 5Centre for Animal Health Studies, Faculty of Natural and Agricultural Sciences, North-West University, Private Bag X2046, Mmabatho 2790, South Africa

**Keywords:** Batswana, biodiversity, endemic, gastrointestinal problems, ethnobotanical survey, indigenous diagnostics, indigenous knowledge, wounds

## Abstract

Ethnoveterinary practices and ethnobotanical knowledge serve as potential therapeutic approaches used to manage and prevent cattle diseases within poor communities in developing nations. Most of the knowledge and practices remain inadequately documented and threatened with extinction in the future. This study aimed to explore the ethnoveterinary practices and knowledge on plants used to treat cattle diseases in two communities of the Ramotshere Moiloa local municipality, South Africa. A semi-structured interview guide, snowball, and purposive technique were used to collect data and recruit 90 participants. Three ethnobotanical indices (informant consensus factor (Fic), use-value (UV), and relative frequency of citation (RFC) were used for quantitative analysis. A total of 64 medicinal plants from 32 families (dominated by Compositae, Fabaceae, and Asparagaceae) were used to treat 27 cattle diseases. The plants with a high frequency of citation and RFC were *Gomphocarpus fruticosus* (75, 0.83), *Opuntia ficus-indica* (74, 0.82), *Schkuhria pinnata* and *Portulaca oleracea* (73, 0.81), *Solanum lichtensteinii* (70, 0.77), and *Senna italica.* In addition, *Schkuhria pinnata* and *Aloe greatheadii* (0.077) had the highest UV. About 28.13% of 64 identified plants were documented as ethnoveterinary medicine for treating cattle ailments, for the first time. The remedies were mainly either prepared as a decoction (52.04%), ground, or prepared as an infusion (16.33%). The plants were administered either orally (69.79%) or topically (30.2%). The ailments with a high frequency of citations were: wounds and constipation (76); arthralgia and retained placenta (69); and lumpy skin disease (68). The categories with the highest number of plants used were gastrointestinal problems (53), skin problems (33), respiratory problems (25), and fertility/reproduction disorders (21). The highest Fic score was cited for tick-borne diseases (1), followed by musculoskeletal systems (Fic = 0.89), and general system infection (Fic = 0.88). The current findings contribute to the documentation and preservation of valuable knowledge from indigenous communities for extensive use. Additionally, ethnoveterinary uses of *Portulaca oleracea*, *Securidaca longipedunculata,* and *Plumbago zeylanica* were recorded for the first time. Further scientific evaluation of the most cited and indigenous/native plants is recommended to establish their therapeutic potential and possible integration into the conventional veterinary sector for the welfare of cattle.

## 1. Introduction

Cattle production plays a key role in the rural economies of developing countries in terms of food security, poverty alleviation, and diverse cultural activities, particularly in rural communities [1,2]. Due to their use as draft animals and their ability to convert low-quality forage into energy-dense muscle and milk, cattle provide a significant source of food and nutrition, much-needed income, and nitrogen-rich manure for replenishing soils and other uses [3,4]. They also fulfil a wide variety of socio-cultural roles. However, cattle in rural areas are often susceptible to various diseases [5]. Changes in population and climate, technology, lifestyles, consumer demands, markets, and other factors are driving rapid change in cattle production. These factors are influencing the way cattle are being produced, improving the livelihoods of people, and sometimes threatening cattle diversity at the local, national, and regional herd levels [6,7,8]. Healthy, well-cared-for, and productive cattle contribute to the sustainable, healthier, and inclusive future livelihood of the communities. The clinical service of the public veterinary service is believed to be inefficient and seen to have minimal effect on animal health [9]. Therefore, maintaining and restoring the health and well-being of the cattle is a critical responsibility for the community members who depend on them.

Farmers and cattle herders in rural communities rely on ethnoveterinary medicine (EVM) as a sustainable alternative to western veterinary practices. Ethnoveterinary medicine encompasses a variety of systems and knowledge of maintaining animal health that is based on beliefs, traditional knowledge, skills, methods, medicinal plants, metaphysics, surgical procedures, technologies, and teachings that are used in healing livestock [10]. The popularity of EVM is often attributed to its ability to improve folk pharmacotherapy which is locally available, economically feasible, accessible, and culturally appropriate [11]. Furthermore, the practice includes a set of empirical observations of the environment and self-management [9]. According to McGaw and Eloff [8], studies in EVM are necessary because plants contain a wide range of phytochemicals. These plants can provide the lead candidates for drug discovery and development of active products, which are useful in managing the health of livestock. In South Africa, the rich and unique flora have been well-utilised in traditional medicine, thereby creating more interest in the potential of medicinal plants [12,13]. As a megadiverse country with a rapidly growing population, the incessant loss of biodiversity justifies the need to document the plant resources, especially the native plants which can be considered as endemic or indigenous [14].

Cattle diseases are major veterinary health problems, which are experienced by livestock farmers in developing countries. Recently, the Conventional Veterinary Services and Drug Resistance reported a rise in the number of cattle diseases that are affecting cattle production [15]. The situation has been compounded by the inaccessibility of veterinary services by rural cattle breeders and the escalating cost of veterinary services. Despite the existing efforts [1,8,16,17,18], there is a paucity of documentation and scientific data regarding the knowledge and practices of ethnoveterinary medicine among different ethnic groups globally.

The majority of indigenous diagnostic and ethnobotanical knowledge methods used in cattle healthcare have been passed down from generation to generation, mainly by word of mouth and apprenticeship [19,20]. Currently, such indigenous knowledge is held by the community’s elders and the limited young members with interest in learning how to use it [21]. Furthermore, EVM is often locally and culturally specific due to differences in disease epidemiology, culture, and biodiversity. Therefore, if not documented, the immense knowledge, skills, and experience accumulated over generations may become extinct in developing countries because of migrations, urbanisation, and technological development [22,23]. Thus, this study explored the indigenous diagnostic and ethnobotanical skills, methods, and processes used to treat cattle diseases and other husbandry indications among Batswana in two communities in the Ramotshere Moiloa local municipality, South Africa.

## 2. Results and Discussion

### 2.1. Socio-Demographic Data of Participants in the Study

A total of 90 community members with ages ranging from 18 to 95 years participated in this study (Table 1). The dominant age group, who constituted about 40% of the participants, were aged 61 and above and are regarded as elders in the communities. Generally, indigenous knowledge on the use of EVM is mostly limited to older people in the communities [24,25,26,27,28]. In the current study, all (100%) of the participants acquired knowledge about indigenous diagnostic skills and ethnobotanical knowledge about diseases from elders. This indicates the relative transmission of indigenous practices from one generation to the next one. The environment and experience of others remain two of the most active means to transmit knowledge about the medicinal values of plants [29,30]. This also indicates that the knowledge is facing a threat, which has a negative impact on the use of ethnoveterinary medicine. Likewise, Giday and Teklehaymanot [31] acknowledged that indigenous knowledge is declining among the younger generation in Africa. However, ethnoveterinary medicine is still prevalent in remote villages of the Ramotshere Moiloa local municipality. The need to preserve the indigenous knowledge, which is at risk of being lost due to the modern lifestyle, remains pertinent [23].

Gender plays a significant part in ethnoveterinary medicine, and the distribution of the participants by gender was 83% male and 17% female. A similar and common pattern indicated that livestock remain mostly in the care of males rather than females [16,32,33]. In Ethiopia, Assefa, and Bahiru [34] indicated that cattle rearing is under the command of males, which influences the imbalanced gender ratio in the practice of ethnoveterinary healthcare. On the other hand, the dominance of females was evidenced in a few studies from countries such as India [35] and China [28].

About 27% of the participants had extensive (more than 40 years) experience in cattle production (Table 1). In the Alaknanda catchment of Uttarakhand in India and the Buyi people of Southwest Guizhou in China, elderly people and male participants were more experienced and had more traditional knowledge of cattle production [28,35]. In the current study, 93% of participants were practicing subsistence farming, whereas a few (7%) engaged in commercial systems. The participants treated the cattle diseases/conditions using medicinal plants (44.4%) or the combination of medicinal plants and conventional medicine (55.6%). Similar results were reported in Eastern Cape, South Africa [36,37]. The use of both methods depended on the availability of funds to procure the conventional medicine, availability of veterinary services, knowledge of indigenous diagnostic methods and plants, value of the animal, and seriousness of the condition [38].

### 2.2. Common Cattle Diseases Treated Using Ethnoveterinary Medicine

Based on the Ruminant Veterinary Association of South Africa [39] and the classification in a previous study [40], the identified cattle diseases were classified into nine categories (Table 2). An inventory of disease conditions identified by the participants was recorded in a generated database with descriptions of disease categories, names in English and Setswana (local name), signs, and symptoms, causes, affected sex and age, and seasonality of outbreak. When compared to the western veterinary medical system, the naming of ailments by indigenous people did not always discriminate between ailments and symptoms of diseases. This is due to the fact that indigenous ailment nomenclature focuses on symptoms, but diseases in western veterinary science are based on aetiological knowledge [9].

The participants identified 27 cattle diseases prevalent in the study area (Table 2). The most often cited ailments were wounds and constipation (76); arthralgia and retained placenta (69); and lumpy skin disease (68). The categories with the highest plant species used were gastrointestinal problems (53), skin problem (33), respiratory problem (25), and fertility/reproduction disorders (21). The informant consensus factor (Fic) is determined by the availability of plants in the study area for ailments treatment. The Fic values in this study varied from 0.56 to 1, with an average of 0.80, indicating a high level of agreement among the participants on the use of plants to treat cattle ailments. Tick-borne diseases had the highest Fic (1), followed by musculoskeletal systems (Fic = 0.89), and general system infection (Fic = 0.88). The high Fic values observed in this study show reasonably reliable knowledge of medicinal plants among the participants. A high Fic value is commonly associated with a few specific plants that have high use reports for treating a single disease category, whereas low values are associated with plants that have almost equal or high UR, implying a lower level of agreement among participants on the use of these plants to treat a specific disease category.

### 2.3. Diagnostic Skills, Treatment Methods, and Endpoint Determination

Participants reported signs and symptoms of cattle diseases/conditions, which they use for diagnosis. Seventy-five distinct clinical signs and symptoms of disease were reported by participants in this study. The most common ones were weight loss, loss of appetite, swelling, weakness and tiredness, breathing problem, distress, restless and discomfort, and blood in the faeces (Table 2). Different clinical signs and symptoms were based on the identified diseases. The current findings suggest a high degree of common perception between ethnoveterinary medicine and conventional veterinary systems. The descriptions of cattle ailments were mostly not like that of the conventional veterinary system, as the participants used signs and symptoms. In some cases, there were some similarities. For example, in terms of tick infestation, participants identified six distinct clinical signs and symptoms corresponding closely to conventional veterinary system concepts of external parasites conditions.

### 2.4. Plants identified as Ethnoveterinary Medicine for Treating Cattle Diseases

In this study, 64 plants were documented as medicine used against cattle diseases (Table 3). The current inventory was higher when compared to those documented in previous studies conducted in the North West Province [25,33,40]. This study reports on new plants that were not documented in earlier studies [25,33]. Particularly, 18 plants (28.13%) were described as ethnoveterinary medicine in treating cattle for the first-time (Table 3). The RFC indicates the local importance of plant species with reference to the participants, who cited the uses of these plants [41]. In the study, the RFC ranged from 0.12 to 0.83. Based on the RFC values, the most cited plant species were *Gomphocarpus fruticosus* (L.) W.T.Aiton (0.83), *Opuntia ficus-indica* (L.) Mill. (0.82), *Schkuhria pinnata* (Lam.) Kuntze ex Thell. and *Portulaca oleracea* L. (0.81), and *Solanum lichtensteinii* Willd. (0.77).

The use-value (UV) is a measure of the types of uses attributed to a particular plant species. In the present study, *Schkuhria pinnata* (Lam.) Kuntze ex Thell., *Senna italica* Mill., and *Aloe greatheadii* Schönland had the highest UV (0.077) followed by *Cleome gynandra* L., *Harpagophytum procumbens* (Burch.) DC. ex Meisn., *Ximenia caffra* Sond. (0.066), and *Ziziphus mucronata* Willd., *Senna italica* Mill., *Portulaca oleracea* L., and *Opuntia ficus-indica* (L) (0.055) (Table 3). The extent to which a species may be employed is determined by its UV; hence, plantss with a high UV are more exploited in the research area to treat more diseases than those with a low UV. Plants with a higher number of use reports (UR) had higher UV, whereas plants with fewer UR by participants had lower UVs, whereas plants with fewer Ui reported by participants had lower UV. Generally, plants that are utilised repeatedly are more likely to be physiologically active [42]. Given that UV and RFC values are dynamic and alter with location and people’s awareness, UV and RFC values may vary from area to area and even within the same study area [43]. Plants with lower UV and RFC values are not necessarily unimportant, but their low values may indicate that the participants are unaware of the uses of these plants and, as a result, that understanding of their use is at risk of not being passed down to future generations, and thus this knowledge may eventually disappear. Some of the documented plants are indigenous to the study area and are well-known to the participants. As a result, their specialised qualities for healing various ailments have become well-known and well-established among the participants. Plants with higher UV and RFC are likely to be good candidates for future research. It will be essential to subject these plants to pharmacological, phytochemical, and biological investigation to establish their therapeutic potential and the potential development of low-cost products [44].

In the current study, the recorded medicinal plants consisted of indigenous (native) and non-endemic plants used against various cattle diseases by the participants of Ramotshere Moloa local municipality. Among the plants, 50 (78.1%) were indigenous/native while 14 (21.8%) were introduced/cultivated/naturalised (Table 3). High levels of usage of these indigenous/native species, particularly those with high use categories, might be of conservation concern if conducted in an unsustainable manner [14]. Thus, there is a great need to discover new biologically active compounds from herbal plants and develop novel drugs. Few studies are available about EVM plants and their constituents with antimicrobial activities [1,8], and these indigenous plants may contain pharmaceutically essential compounds. To further understand the uniqueness of this EVM, a more in-depth study of how these indigenous plants is used and selected, as well as a comparison study with other sites/locations within South Africa is required. In addition, a closer look at the local conservation status is required to build a sustainable use plan for these valuable plant resources [23].

### 2.5. Therapeutic Uses of Combined Medicinal Plants

Participants in the current study reported nine (14.1%) plants from the inventory that had numerous indications (uses), as poly-plant remedies. These results reflect the diversity of ethnic knowledge and heterogeneity in cultural practices. For example, participants reported using a combination of the leaves of *Artemisia afra* Jacq. ex Willd., *Mentha aquatica* L., *Dicoma macrocephala* DC., and *Lippia scaberrima* Sond., and mixing them with donkey milk to cure cough, intestinal worms, and joints pain. A decoction of *Drimia sanguinea* (Schinz) Jessop bulb and the roots of *Elephantorrhiza burkei* Benth. and *Senna italica* Mill. was administered orally to treat intestinal worms. To treat constipation, a decoction of the roots of *Elephantorrhiza burkei* Benth., *Peltrophorum africanum* Sond., and *Jatropha zeyheri* Sond. is administered orally. The potency of using a combination of different plants or plant parts increased compared to using a single plant to cure a disease is well-recognised [25,40,45]. Validation and transmission of this knowledge to livestock raising farmers all over the world so that they know the best plant material near them for the specific ailment will benefit people not only in impoverished nations but also in the developed world [46]. The use of two or more plants exemplifies the notion of synergy, which highlights that the combination of plants might result in higher therapeutic efficacy [32,47].

### 2.6. Plant Families Used to Treat Cattle Diseases

In terms of family diversity, 32 families were used to treat and manage cattle diseases in the study area (Figure 1). The families with the largest number of plant species used to treat cattle were Compositae with seven species and Fabaceae and Asparagaceae (five species). Compositae and Fabaceae are the most widely used families in ethnoveterinary studies [18,48,49,50]. Similar studies have been reported where participants mostly used the members of Compositae for the preparation of EVM for the treatment of different livestock diseases [40,51,52]. Furthermore, the widespread use of plants from these dominant families might be attributed to strong traditional beliefs, availability, ease of harvesting, and storage, as well as the evidence of bioactive compounds with therapeutic effect against cattle ailments. However, the trend for plant families utilised to cure cattle ailments in the selected communities differs from those used in other locations in South Africa [33,53].

### 2.7. Distribution of Plant Parts Used to Treat Cattle Diseases

In the study area, different plant parts were used for the preparation of remedies for treating cattle diseases (Figure 2). The most frequently used plant parts were the leaves (34.55%), roots (25.45%), and whole plant (19.09%). The preference of leaves in treating cattle ailments is due to their easy availability, easy harvesting, and simplicity in remedy preparation. Leaves are the storage site of diverse pools of phytochemicals, the renewable parts of plants, and for a conservation perspective, their collection may not result in the fatality of the mother plants [54]. A similar trend whereby the leaves were the dominant plant part used in medicine preparation for treating cattle diseases was reported in other studies [51,55,56,57,58]. However, in some cases, the roots were identified as the commonly used plant parts [9,33,40]. Roots were the second most commonly utilised plant part, which might be attributed to the fact that roots remain in the soil and are easily accessible even during extended dry seasons in arid and semi-arid environments [59]. However, this is frequently not suggested because it is harmful and unsustainable, putting plant species at risk of extinction [60].

### 2.8. Method of Preparation and Route of Administration of Medicinal Plants Used to Treat Cattle Diseases

As depicted in Figure 3, the herbal remedies were mainly prepared as decoction (52.04%), ground, and infusion (16.33%). Other ways of preparation were maceration, poulticing, and burning which cumulatively accounted for 31.63%. Decoction is the process of boiling plant components in water and then allowing the liquid to cool before administering. Decoction is a popularpreparation methodthat has been mentioned in several different research studies [48,61,62]. The preparation method differs from other study areas including Karamoja in Uganda [63], the Mana Angetu district of south-eastern Ethiopia [64], and Yalo Woreda in the Afar regional state, Ethiopia [65], where crushing and pounding were the most common methods. The widespread usage of decoction might be attributed to the fact that boiling can accelerate biological processes, resulting in the greater availability of several active compounds [66]. However, preparation procedures vary based on the type of sickness being treated, the location of the condition, and the medicinal components to be extracted [67].

Furthermore, the local communities use a variety of methods to administer the plants when treating cattle diseases (Figure 4). The route of administration for the plants was oral (69.79%) and topical (30.2%). Oral administration is a simple and non-invasive form of systemic treatment. The route allows for the rapid absorption and distribution of the prepared medicines and allowing for sufficient curative power to be delivered [68]. Across many African cultures, oral administration of medicinal plants is the most common route used to treat disease in cattle, as this ensures fast and direct interaction with different plant compounds at the site of action [69,70,71].

## 3. Materials and Methods

### 3.1. Study Area

This study was conducted in the Gopane (25.3175° S, 25.8231° E) and Dinokana (25.456° S, 25.8799° E) villages of the Ramotshere Moiloa local municipality (25.5623° S, 26.1001° E) located in the Ngaka Modiri Molema district municipality, North West Province of South Africa (Appendix A and Figure 5). Dinokana and Gopane are bordered to the north by Botswana, to the east by Moses Kotane and Kgetleng River local municipalities, and to the south by Ditsobotla and Mafikeng local municipalities [72]. The two communities are dominantly rural area under the leadership of traditional leader and municipality councillors. The area is rich in floras with potential diverse applications [73]. The total surface of both communities is 104,882 km^2^. Livestock and agricultural productions provide a significant contribution to the rural economy in the study region, and most rural farming systems transport and generate money both directly and indirectly.

The households are active in agricultural activities such as livestock (16,443) and vegetable (1110) production. Particularly, cattle production, consisting of 11,892 households, remains the highest agricultural activity in these communities. The annual income category of agricultural household heads starts from R1 to above R1 228,800 [72], and according to the Ruminant Veterinary Association of South Africa [39], the most prevalent animal diseases in the study are internal parasites, external parasites, tick-borne diseases, insect transmittable diseases, venereal diseases, bacterial diseases, and protozoal diseases. The two communities were selected due to the rich plant biodiversity which serves as an important medicinal resource [74]. Increased population growth, cultural changes such as the rapid shift toward allopathic medicine, and the spread of modern education contribute to the destruction of medicinal plant habitats and the increasing loss of indigenous knowledge due to changes in community inhabitants. Previously, Van der Merwe, Swan, and Botha [33] documented the ethnoveterinary medicine knowledge of the Madikwe community and Ndou [40] focused on the EVM in Mahikeng, whereas another study focused on the medicinal plants used for retained placenta [25].

### 3.2. Ethnobotanical Survey

An ethnobotanical field survey was conducted from August 2019 to October 2020 (Spring until Summer) in the Dinokana and Gopane villages of the Ramotshere Moiloa local municipality, South Africa (Figure 5). Snowballing was used to recruit and screen eligible participants [75,76]. Ninety participants (83% were male and 17% female) were purposively selected to participate in the study. The age of participants ranged from 18 to 95 and the participants consisted of indigenous knowledge holders, farmers, and cattle herders. The experience and knowledge of participants on the theme of the study, and their interest in participating, were applied as the inclusion criteria [77].

A face-to-face interview using a semi-structured interview guide prepared in English and translated to Setswana (local language) was used to collect data, after presenting the purpose of the study to the participants, and data was subsequently translated to English. The semi-structured interview guide yielded insightful knowledge for the researcher to develop and generate a rich understanding of the knowledge and skills related to ethnoveterinary [78]. The data collection questionnaire was divided into three sections to obtain required information. Two phases were followed to collect data. The first phase was interviews, and the second involved a field walk and the collection of plants. Following the Alexiades and Sheldon [79] technique, responses of participants that contradicted each other were not considered for analysis. The data generated from individual interviews were cross-checked with other participants in the same villages to obtain reliable information in the study area [80].

The Faculty of Natural and Agricultural Sciences Research Ethics Committee (FNASREC) at the North-West University reviewed and approved the study (Ethics approval number: NWU-01228-19-S9). Traditional authorities in the local municipality granted permission and access to conduct the study in the communities. Prospective participants were approached to seek their consent to participate in the study following detailed and clear explanation on the purpose of the research. The North West Department of Rural, Environmental, and Agricultural Development (NW-READ) granted authorisation for plants collection in the two villages (Permit number HQ 26/01/18-006 NW).

Field observations were conducted in study areas to collect the plants mentioned during interviews. Plants were identified by participants and collected by researchers during field walks. Plants used to treat cattle diseases were collected using standard procedures/techniques [81]. Voucher specimens for the plants were prepared and deposited in the SD Phalatse (UNWH) and AP Goossens Herbarium (PUC) at the North-West University. The nomenclature of all the collected plants was verified using The World Flora Online (http://www.worldfloraonline.org/, accessed on 22 March 2022).

### 3.3. Data Analysis

Thematic content and ethnobotanical indices were used to analyse the data collected on indigenous diagnostic skills and ethnobotanical knowledge provided by participants. Thematic content analysis was used to analyse qualitative data [82]. Following the interviews, the data was transcribed and verified for coherence and saturation. The information from various participants was compared to each other to uncover trends and themes. The emergent themes were linked to data sections with corresponding codes such as participant socio-demographic information, frequently identified cattle ailments, diagnostic procedure, and medicinal plant usage process. When no new data, codes, or themes came from the material, it was considered that saturation had been reached. The ethnobotanical knowledge data were analysed using informant consensus factor (Fic), use-value (UV), and relative frequency of citation (RFC) as described below:

Informant consensus factor (Fic): relevant for the categories of diseases to identify the agreement of participants on the reported cures for the group of diseases [83], which was calculated as follows:$$\mathrm{Fic}=\frac{\mathrm{Nur}-\mathrm{Nt}}{\left(\mathrm{Nur}-1\right)}$$
where Nur denotes the number of usage reports for a certain ailments category and Nt denotes the number of plants listed for the treatment of that ailment category.

Use value (UV): denotes the relative significance of species recognized locally [84]. The UV was used to identify the plants with the highest utilisation translating to the most frequently mentioned in the treatment of cattle disease [85]. It was calculated as follows:$$\mathrm{UV}=\frac{\mathrm{Ui}}{N}$$
where Ui: is the number of uses stated by each participant for a specific species and N: denotes the total number of participants. If a plant secures a high UV score, that indicates that there are many use reports for that plant, whereas a low score indicates fewer use reports cited by the participants.

Relative frequency of citation (RFC): as described by Tardío and Pardo-de-Santayana [86], this measures the agreement among participants on the reported plants. This index calculates the local relevance of each species by dividing the number of participants who mention the species’ use, also known as frequency of citation (FC), by the number of target participants included in the study (N). The RFC index was calculated using the following formula:$$\mathrm{RFC}=\frac{\mathrm{FC}}{N}\;(0<\mathrm{RFC}<1)\;$$
where FC is the number of participants who reported using a certain species and N denotes the total number of participants in the research. The factor has a value range of 0 to 1, with a high value indicating a high rate of participant consensus.

## 4. Conclusions

The selected communities are primarily rural in nature, and cattle farmers are exploring their biodiversity and indigenous knowledge practices for meeting the animal health needs and productivity. Based on the current findings, an inventory of 64 medicinal plants from 32 families with a specific indigenous/native rate of 78.1% used to treat 27 cattle ailments from nine categories was documented, with 18 new plants. Three diagnostic skills, 75 distinct clinical signs and symptoms of disease, and two endpoint determinations were reported to understand cattle diseases. Leaves as a plant part, decoction as a preparation method, and oral as an administration route were found to be the most frequently used systems in treating cattle diseases. The plants were prepared as monotherapy and combination. Even though the research area in the Ramotshere Moiloa local municipality was shown to be rich in medicinal plant variety, efforts to study the plants and the indigenous knowledge connected with them are currently limited. To avert additional losses, local communities and responsible entities must conserve therapeutic plants. Furthermore, plants with a high potential based on the applicable ethnobotanical indices should be selected for additional research, such as phytochemical analysis and pharmacological and toxicological studies.

## Figures and Tables

**Figure 1 plants-11-01784-f001:**
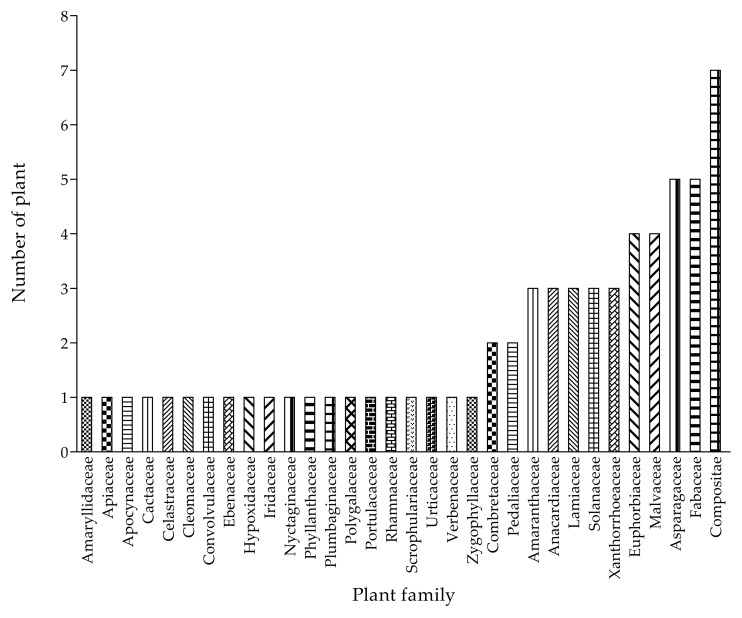
Frequency of the 32 plant families used in the treatment of cattle diseases in the Ramotshere Moiloa local municipality, North West Province, South Africa.

**Figure 2 plants-11-01784-f002:**
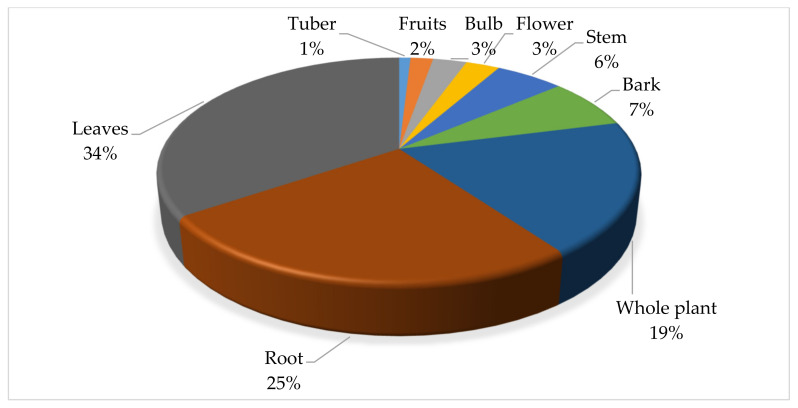
Distribution (%) of plant parts value used in the treatment of cattle diseases in Ramotshere Moiloa local municipality, North West Province, South Africa. (*n* = 110).

**Figure 3 plants-11-01784-f003:**
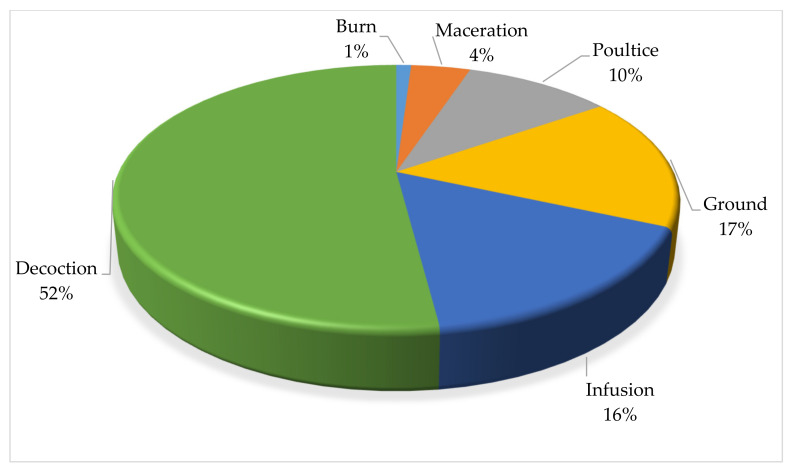
Distribution (%) of preparation methods for plants used in the treatment of cattle diseases in Ramotshere Moiloa local municipality, North West Province, South Africa (n = 98).

**Figure 4 plants-11-01784-f004:**
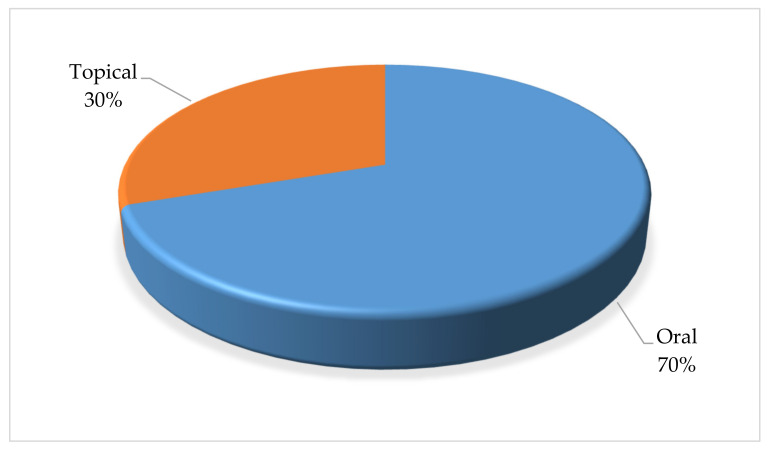
Distribution (%) pattern for the administration of plants used in the treatment of cattle diseases in Ramotshere Moiloa local Municipality, North West Province, South Africa (n = 96).

**Figure 5 plants-11-01784-f005:**
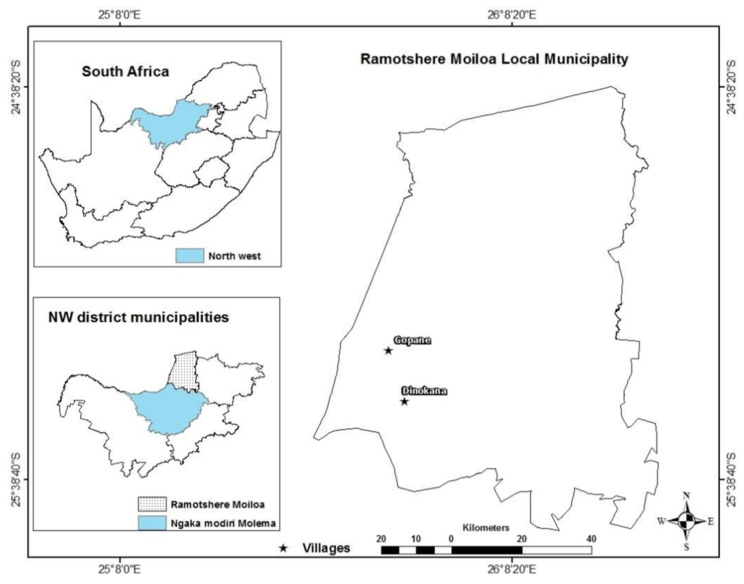
Location of the Dinokana and Gopane villages in the Ramotshere Moiloa local municipality, North West Province, South Africa.

**Table 1 plants-11-01784-t001:** Demographic information of participants engaged in the study (*n* = 90).

Characteristic of Participants	Category	Frequency	Distribution (%)
1. Age group (years)	18–30	12	13
	31–40	15	17
	41–50	12	13
	51–60	15	17
	61 and above	36	40
2. Gender	Male	75	83
	Female	15	17
3. Experience in cattle production (years)	5–10	12	13
	11–15	9	10
	16–20	15	17
	21–30	15	17
	31–40	15	17
	41 and above	24	27
4. Form of cattle production	Subsistence	84	93
	Commercial	6	7
5. Method of animal treatment	Medicinal plants only	40	44.4
	Combination of medicinal plants and conventional medicine	50	55.6

**Table 2 plants-11-01784-t002:** Distribution of cattle diseases, causes, seasonality, sex, signs, and symptoms reported by participants in the study area. Nur denotes the number of usage reports for a certain disease category, whereas Nt denotes the variety of plants cited for the treatment of that specific ailment category, Fic = informant consensus factor.

Disease Categories	Diseases	Local Name	Citation	Nur	Nt	Fic	Signs and Symptoms	Causes	Sex	Seasonality of Diseases
Gastrointestinal problems				120	53	0.56				
	Dysentery	Letsholo le lehibidu	4		1		Same as diarrhoea, the difference is the colour, dysentery faeces are reddish	Unspecified	Both	All seasons
	Diarrhoea	Letsholo	32		23		Watery faeces and pass them often, faeces are unusual colour and smell, sometimes blood in the faeces, back legs are dirty with faeces, become weak, tired, loss of appetite, appearing distressed and restless, weight loss, eyes sink into the head	Eating dry or green grass Feeding on grass containing worms (Kgorosane)	Both	Summer Spring
	Bile reflux	Gala	8		1		Gall bladder is large and full of brown/green liquid, loss of appetite, loss of weight, dry nose, drowsy, and brownish sclera	Green grass	Both	Summer
	Constipation	Go shokega mala	76		28		Passing dry, small, and hard faeces, no sign of faeces passed in the night, little dark urine, looking distressed when it passes faeces or urine, loss of appetite	Low intake of fluids	Both	All seasons
							Tilt the affected ear downwards and may roll or lean to the affected side.	Unknown	Both	All seasons
General system infection				63	8	0.88	Weight loss, restless, loss of appetite, weak and tired, and has become very thin.	Not eating well	Both	All seasons
	Ear pain	Bolwetsi jwa ditsebe	25		3		Stiff shoulders, loss of weight, loss of appetite, feeling bubbles under the skin, weak and tired, swelling legs, high fever	Change of seasons Insects in the grass Heat	Both	Autumn and Winter
	Malnutrition	Sekwepe	10		1		Saliva comes from the mouth, high fever, loss of appetite, blood in the urine, faeces, or milk, swelling under the jaw, neck, chest, and abdomen, become sick suddenly, difficulty breathing, collapse and die, dark blood comes from the mouth, nose, or anus when it is dead	Insects in the soil Too much blood	Both	All seasons
	Blackquarter	Serotswana	5		1		Difficulty in breathing, looking distressed and restless, lame, and cannot walk normally	Unknown	Both	All seasons
	Anthrax	Lebete	5		1		Painful shoulders, isolation, stiff shoulders, loss of appetite, losing weight, walking with head down looking like it is carrying a heavy load, breathing heavy	Kicking stones Heat	Both	All seasons
	Heart problems	Bolowetsi jwa pelo	10		1		Tilt the affected ear downwards and may roll or lean to the affected side.	Unknown	Both	All seasons
	Aphosphorisis	Mokokomalo/Magetla/lamsiekte	8		1		Weight loss, restless, loss of appetite, weak and tired, and has become very thin	Not eating well	Both	All seasons
Fertility/Reproduction disorders				81	21	0.75				
	Pain afterbirth	Ditlhabi tsa morago ga pelegi	1		1		Redness around vulva, swelling, warmth	Dystocia process of pulling the dead calf	Female	Summer–Autumn
	Retained placenta	Motlhana	69		10		Placenta hanging from the vulva for a long time, smell bad if stay for a long time, restless	Unknown	Female	Summer–Autumn
	Dystocia	Go farelwa	11		10		Breathing fast and heavily, moves from time to time/restless, unable to give birth	Not eating well Not enough water Running	Female	Summer–Autumn
Skin problem				190	33	0.83				
	Foot rot	Dintho tsa dikgato	5		1		Swelling between the two hooves and legs, limping and lifting the affected leg, the flesh between the two claws that looks damaged, hard, or cut, wound or cut is smelling bad	Cow walking in the rainwater	Both	Summer
	Abscess	Ditlhagala/Knopsik	41		14		Hard, hot, and painful swelling on the body but often just under the skin, that becomes soft, grey/white/green/yellow fluid comes out when it bursts, foot smells or is hot and painful, loss of appetite	Rainwater Dirty blood	Both	Summer Spring Autumn
	Lumpy skin disease	Ditompola	68		1		Lots of saliva, clear discharge comes from the eyes and grey/white mucus from the nose, weak and tired, hard lumps appear on the body, hair around and on the lumps stands up, loss of appetite	Change of seasons	Both	Summer
	Wounds	Dintho	76		17		A cut on any outside part of the body, loss of appetite	Cut from objects	Both	All seasons
Internal and external parasites				91	12	0.87				
	Worms	Dibokwana	50		10		Weight loss, loss of appetite, skin rough, swelling throat, breathing problem, bloody diarrhoea with mucus, dehydration	Bad food.	Both	All seasons
	Ticks	Dikgoa	41		2		Licking and rubbing at the bite sites, lack of energy, loss of condition, tick sores and ulceration, pale parts around the eyes due to anaemia	Exposed to the cattle with ticks	Both	All seasons
Musculoskeletal systems				124	14	0.89				
	Muscle pain	Ditlhabi	10		4		Facial expressions	Diseases	Both	All seasons
	Arthralgia	Ditlhabi tsa malokololo	69		8		Facial expressions	Diseases	Both	All seasons
	Fracture	Go robega	45		2		Lame and limp, hold a broken leg off the ground and does not put any weight on that leg, when pressed under the foot it feels the pain, and swelling around the broken leg when touched you can feel broken ends of the bone, look distressed and restless, you can hear a grinding noise of the broken bones	Objects Kicking stones Losing step when running	Both	All seasons
Respiratory problems				100	25	0.76				
	Cough	Go gotlhola	42		23		Lost weight, looks weak and tired, cough most of the time, loses appetite, sneeze, white/yellow mucus coming out of the nose, the problem with breathing	Change of seasons Cold	Both	Summer– Winter
	Lung diseases	Lamsik/makgwafo a botlhoko	9		1		Loss of appetite, appearing distressed and restless, weight loss, has become thin, breathing problems	Heat	Mostly Female	Spring
	Lung congestion	Galbater/metsi mo mafatlheng	45		1		Not eating as much as normal, weight loss, looking distressed and restless, breathing problems	Eating dry grass	Both	Autumn– Winter
Tick-borne				5	1	1				
	Heartwater	Bolwetsi jwa metsi mo pelo	5		1		Diarrhoea and signs of bleeding from the anus, move-in violent and convulsive, blood in faeces, grinding their teeth, lifting their legs up very high when they walk, loss of appetite, the sac around the heart, chest, and abdomen is full of fluid, collapse and die	Ticks	Both	All seasons
Eye problem				55	10	0.83				
	Eye infection	Botlhoko jwa matlho	55		10		Walks into objects, quick movement when an object comes towards the eye, the eye is red, and the eyelids are swollen, it blinks a lot and avoids bright sunlight, watery/yellowish discharge that smell bad comes out of an eye, look distressed and restless, the skin under the eyelids turn red, loss of appetite	Sharp objects pricking, insects, flies, or dust	Both	All seasons

**Table 3 plants-11-01784-t003:** Medicinal plants used to treat various cattle ailments in the Ramotshere Moiloa local municipality (FC = number of participants who reported using a certain species, UV = use-value, Ui = the number of uses stated by each participant for a specific species, RFC = relative frequency of citation, * = plants possibly identified for the first time used for treating cattle diseases as no records were found). The nomenclature of all the collected plant species has been updated using The World Flora Online (http://www.worldfloraonline.org/, accessed on 22 March 2022). ^I^ Indigenous and ^$^ introduced/cultivated/naturalised plants were based on Plants of Southern Africa (POSA) (http://posa.sanbi.org/, accessed on 16 June 2022) and the African Plant Database (https://africanplantdatabase.ch/, accessed on 22 June 2022).

Scientific Name Voucher Specimen Number	Local Name	Family	FC	Ui	UV	RFC	Plant Part(s)	Disease/ Ailments Treated	Mode of Preparation	Administration Route
^I^*Acrotome* *inflata* Benth. MVCHA 01	Mogato	Lamiaceae	15	2	0.022	0.16	Leaves	Cough	Decoction	Oral
							Whole plant	Wounds	Burn	Topical
^I^*Aloe greatheadii* Schönland MVCHA 02	Kgopane e nyane	Xanthorrhoeaceae	20	7	0.077	0.22	Leaves	Constipation, diarrhoea, retained placenta	Decoction or infusion	Oral
								Ticks, abscesses. Wounds, muscle pain	Ground	Topical
^$^*Aloe vera* (L.) Burm.f. MVCHA 03	Kgopane ya thaba	Xanthorrhoeaceae	67	3	0.033	0.74	Leaves	Abscess, wounds	Ground	Topical
								Retained placenta, diarrhoea	Decoction or infusion	Oral
^$^** Amaranthus cruentus* L. MVCHA 05	Modinakana	Amaranthaceae	12	1	0.011	0.13	Whole plant	constipation	Ground	Oral
^$^*Amaranthus spinosus* L. MVCHA 04	Setlepetlepe	Amaranthaceae	12	3	0.033	0.13	Roots and leaves	Abscess, wounds	Poultice	Topical
							Whole plant	Ear pain	Ground	Topical
^I^*Aptosimum elongatum* (Hiern) Engl. MVCHA 06	Ditantanyane	Scrophulariaceae	11	1	0.011	0.12	Whole plant	Arthralgia	Decoction	Oral
^I^*Artemisia afra* Jacq. ex Willd. MVCHA 07	Lengana	Compositae	69	4	0.044	0.76	Leaves	Cough, intestinal worms, arthralgia	Decoction	Oral
								Ear pain	Ground	Topical
^I^** Asparagus africanus* Lam. (*Asparagus cooperi* Baker) MVCHA08	Thokabotswaro	Asparagaceae	15	1	0.011	0.16	Roots and stems	Malnutrition	Infusion	Oral
^I^*Asparagus laricinus* Burch. MVCHA 09	Lesitwane	Asparagaceae	20	1	0.011	0.22	Whole plant	Muscle pain	Decoction	Oral
^I^*Asparagus suaveolens* Burch. MVCHA 010	Motantanyane	Asparagaceae	15	1	0.011	0.16	Whole plant	Dystocia	Decoction	Oral
^I^** Babiana hypogea* Burch. MVCHA 011	Thuge	Iridaceae	20	2	0.022	0.22	Leaves	Abscess, muscle pain	Infusion	Oral
^$^** Boerhavia diffusa* L. MVCHA 018	Moetapele	Nyctaginaceae	45	3	0.033	0.5	Leaves and stems	Eye infection, abscess, wounds	Decoction	Topical
^I^*Boophone disticha* (L.f.) Herb. MVCHA 012	Lesoma/Mathubadudifala	Amaryllidaceae	45	1	0.011	0.5	Leaves, roots, and bulb	Constipation	Decoction	Oral
^I^*Bulbine abyssinica* A.Rich. MVCHA 013	Makgabenyane	Xanthorrhoeaceae	30	3	0.033	0.33	Leaves	Abscess, wounds	Grounded	Topical
^I^*Cassine transvaalensis* (Burtt Davy) Codd [*Elaeondendron transvaalense* (Burtt Davy) R.H. Archer] MVCHA 014	Mojelemane	Celastraceae	12	1	0.011	0.13	Barks	Diarrhoea	Decoction	Oral
^I^** Centella asiatica* (L.) Urb. MVCHA 015	Setimamolelo	Apiaceae	12	4	0.044	0.13	Leaves	Wound, abscess, eye infection	Poultice	Topical
							Whole plant	Diarrhoea	Decoction	Oral
^I^** Cleome gynandra* L. MVCHA 016	Rothwe	Cleomaceae	35	6	0.066	0.38	Flower	Eye infection, ear problem	Grounded	Topical
							leaves	Wounds	Grounded	Topical
							Roots and leaves	Cough, constipation, intestinal worms	Infusion or decoction	Oral
^I^*Combretum hereroense* Schinz MVCHA 017	Tsholakhudu	Combretaceae	25	4	0.044	0.27	Leaves	Cough, pains, dysentery, constipation	Decoction	Oral
^I^*Croton gratissimus* Burch. MVCHA 019	Moologa	Euphorbiaceae	22	2	0.022	0.24	Flower	Eye infection, ear problem	Grounded	Topical
^I^*Dicerocaryum senecioides* (Klotzsch) Abels (*Sesamum senecioides* (Klotzsch) Bung & Christenh. MVCHA 020	Tshetlho ya mamitlwa a mabedi	Pedaliaceae	57	3	0.033	0.63	Leaves	Blackquarter	Poultice	Topical
							Whole plant	Retained placenta, dystocia	Poultice and infusion	Oral
^I^*Dichrostachys cinerea* (L.) Wight & Arn. MVCHA 021	Moselesele	Fabaceae	23	4	0.044	0.25	Barks	Retained placenta, dystocia, fracture, arthralgia	Poultice	Topical
^I^*Dicoma macrocephala* DC. MVCHA 022	Tlhonya	Compositae	34	1	0.011	0.37	Roots	Diarrhoea	Infusion	Oral
^I^*Drimia sanguinea* (Schinz) Jessop MVCHA 023	Sekaname	Hyacinthaceae	45	3	0.033	0.5	Bulb	Retained placenta, intestinal worms, constipation	Infusion	Oral
^$^** Dysphania ambrosioides* (L.) Mosyakin & Clemants MVCHA 024	Tlhatlhabadimo	Amaranthaceae	56	2	0.022	0.62	Whole plant	Cough, constipation	Infusion	Oral
^I^*Elephantorrhiza burkei* Benth. MVCHA 026	Mositsane	Fabaceae	68	4	0.044	0.75	Roots	Cough, constipation, retained placenta	Decoction	Oral
								Diarrhoea	Ground	Topical
							Barks	Cough	Decoction	Oral
^I^*Euclea undulata* Thunb. MVCHA 070	Morobe	Ebenaceae	67	6	0.066	0.74	Leaves	Wounds	Poultice	Topical
							Bark and roots	Diarrhoea, arthralgia	Decoction	Oral
							Roots	Cough, constipation, retained placenta	Decoction	Oral
^$^** Euphorbia balbisii* Boiss (*Euphorbia serpens* Kunth) MVCHA 027	Lwetsane	Euphorbiaceae	12	2	0.022	0.13	Leaves and roots	Diarrhoea, intestinal worms	Decoction	Oral
^I^*Gomphocarpus fruticosus* (L.) W.T.Aiton MVCHA 028	Motimola/sebogamaswi	Apocynaceae	75	4	0.044	0.83	Whole plant	Constipation, retained placenta, cough, bile reflux	Infusion	Oral
^I^*Grewia flava* DC. MVCHA 029	Moretlwa	Malvaceae	23	2	0.022	0.25	Roots	Diarrhoea, dystocia	Infusion	Oral
^I^** Grewia flavescens* Juss. MVCHA 030	Motsotsojane	Malvaceae	34	3	0.033	0.37	Leaves	Pain, wounds, diarrhoea	Infusion	Oral
^I^*Harpagophytum procumbens* (Burch.) DC. ex Meisn. MVCHA 031	Lematla/Sengaparile	Pedaliaceae	45	6	0.066	0.5	Tuber	Dystocia, pain after birth	Decoction	Oral
								Abscess, fracture, muscle pain	Ground	Topical
							Roots, leaves, and fruit	Retained placenta	Decoction	Oral
^I^*Helichrysum candolleanum* H.Buek MVCHA 032	Phateyangaka	Compositae	56	1	0.011	0.62	Roots, leaves and fruit	Retained placenta	Decoction	Oral
^I^*Hypoxis hemerocallidea* Fisch., C.A. Mey. & Avé-Lall. MVCHA 034	Maledu/Tshuku ya poo	Hypoxidaceae	69	4	0.044	0.76	Whole plant	Cough, dystocia, arthralgia, constipation	Decoction	Oral
^I^*Jatropha zeyheri* Sond. MVCHA 035	Seswagadi	Euphorbiaceae	67	3	0.033	0.74	Root	Eye infections, constipation	Maceration	Topical
								Retained placenta	Decoction	Topical
^I^*Kleinia longiflora* DC. MVCHA 036	Mosimama	Compositae	65	1	0.011	0.72	Whole plant	Eye infection	Poultice	Topical
^I^** Lippia scaberrima* Sond. MVCHA 037	Mosukutswane	Verbenaceae	54	1	0.011	0.6	Leaves	Cough	Decoction	Oral
^$^** Malva neglecta* Wallr. MVCHA 038	Tikamotse	Malvaceae	43	4	0.044	0.47	Leaves and flowers	Constipation, wounds, abscess, cough	Decoction	Oral
^$^*Malvastrum coromandelianum* (L.) Garcke MVCHA 054	Thobega	Malvaceae	12	4	0.044	0.13	Leaves	Diarrhoea, abscess, wounds, ear pain	Decoction	Oral
^I^** Mentha aquatica* L. MVCHA 039	Kgobedimetsing	Lamiaceae	32	1	0.011	0.35	Leaves	Cough	Decoction	Oral
^$^*Opuntia ficus-indica* (L.) Mill. MVCHA 040	Toorofeye	Cactaceae	74	5	0.055	0.82	Leaves and stem	Diarrhoea, constipation, eye infections	Decoction	Oral
							Flower	Retained placenta, abscess	Ground	Topical
							Bulb	Retained placenta	Maceration	Topical
^I^*Ozoroa paniculosa* (Sond.) R. Fern. & A. Fern. MVCHA 042	Monokana	Anacardiaceae	24	2	0.022	0.26	Roots	Cough, muscle pain	Decoction	Oral
^I^*Peltophorum africanum* Sond. MVCHA 043	Mosetlha	Fabaceae	35	5	0.055	0.38	Roots	Wounds, muscle pain	Decoction	Oral
							Stem and root	Diarrhoea, constipation	Decoction	Oral
							Root, Leaves, and bark	Intestinal worms	Decoction	Oral
^I^** Phyllanthus maderaspatensis* L. MVCHA 044	Mositwane	Phyllanthaceae	30	3	0.033	0.33	Whole plant	Eye infection	Ground	Topical
							Roots	Constipation, diarrhoea	Decoction	Oral
^I^** Plumbago zeylanica* L. MVCHA 045	Masegomabe	Plumbaginaceae	57	2	0.022	0.63	Whole plant	Cough, intestinal worms	Decoction	Oral
^$^** Portulaca oleracea* L. MVCHA 046	Selele	Portulacaceae	73	5	0.055	0.81	Whole plant	Constipation, eye infection, muscle pain, wounds, intestinal worms	Decoction	Oral
^I^*Pouzolzia mixta* Solms MVCHA 047	Mongololo	Urticaceae	68	3	0.033	0.75	Roots and leaves	Retained placenta	Maceration	Oral
							Roots	Diarrhoea	Decoction	Oral
								Constipation	Infusion	Oral
^$^*Ricinus communis* L. MVCHA 048	Mokhura	Euphorbiaceae	13	2	0.022	0.14	Leaves	Constipation, eye infection	Infusion	Oral
^I^** Sansevieria hyacinthoides* (L.) Druce [*Dracaena hyacinthoides* (L.) Mabb.] MVCHA 049	Moshokelatsebe	Ruscaceae	24	2	0.022	0.26	Leaves	Retained placenta	Poultice	Topical
							Whole plant	Cough	Decoction	Oral
^$^*Schkuhria pinnata* (Lam.) Kuntze ex Thell. MVCHA 050	Santlhoko	Compositae	73	7	0.077	0.81	Whole plant	Eye infection, wounds, abscess	Ground	Topical
								Diarrhoea, constipation, heartwater, lung congestion	Decoction	Oral
^I^*Searsia lancea* (L.f.) F.A.Barkley MVCHA 051	Moshabela	Anacardiaceae	30	3	0.033	0.33	Roots, leaves, and stem	Abscess	Poultice	Roots, leaves and stem
							Roots	Constipation, diarrhoea	Infusion	Roots
^I^*Searsia pyroides* (Burch.) Moffett MVCHA 052	Bohitlha	Anacardiaceae	57	6	0.066	0.63	Leaves	Cough	Decoction	Oral
							Leaves	Cough, dystocia, constipation, diarrhoea, intestinal worms, arthralgia	Decoction	Oral
^$^** Securidaca longipedunculata* Fresen. MVCHA 053	Mmaba	Polygalaceae	68	4	0.044	0.75	Roots	Cough, Dystocia, constipation, muscle pain	Ground	Topical
^I^*Senecio consanguineus* DC. MVCHA 033		Compositae	67	3	0.033	0.74	Whole plant	Cough, wounds, constipation	Decoction	Oral
^I^*Senna italica* Mill. MVCHA 055	Sebetebete	Fabaceae	55	7	0.077	0.61	Leaves	Constipation, abscess, anthrax, aphosphorosis, lung diseases	Decoction	Oral
							Bark	Pain, diarrhoea, constipation	Decoction	Oral
^I^*Solanum lichtensteinii* Willd. MVCHA 059	Tolwane	Solanaceae	70	1	0.011	0.77	Whole plant	Ticks	Poultice	Topical
^$^*Solanum nigrum* L. MVCHA 060	Makgonatsotlhe	Solanaceae	33	1	0.011	0.36	Roots	Intestinal worms	infusion	Oral
^I^*Solanum campylacanthum* A. Rich. (*Solanum panduriforme* Dunal) MVCHA 071	Tolwane enyane	Solanaceae	44	2	0.022	0.48	Roots	Diarrhoea	Infusion	Oral
							Leaves	Eye infection	Maceration	Topical
^I^*Tarchonanthus camphoratus* L. MVCHA 061	Mohatlha	Compositae	33	1	0.011	0.36	Roots	Intestinal worms	infusion	Oral
^I^*Terminalia sericea* Burch. ex DC. MVCHA 062	Mogonono	Combretaceae	44	1	0.011	0.48	Leaves and stem	Cough	Decoction	Oral
^I^*Teucrium sessiliforum* Benth. (*Teucrium trifidum* Retz.) MVCHA 063	Lethe la noga	Lamiaceae	55	3	0.033	0.61	Leaves and roots	Cough, diarrhoea, constipation	Decoction	Oral
^I^*Tribulus terrestris* L. MVCHA 064	Tshetlho	Zygophyllaceae	66	5	0.055	0.73	Leaves	Arthralgia	Ground	Oral
							Whole plant	Dystocia, retained placenta	Decoction	Oral
						.		Constipation	Decoction	Oral
^I^** Turbina oblongata* A. Meeuse (*Ipomoea oblongata* E.Mey ex Choisy) MVCHA 065	Mokatelo	Convolvulaceae	11	4	0.044	0.12	Roots	Cough, wounds, muscle pain, diarrhoea	Decoction	Oral
^I^*Vachellia karroo* (Hayne) Banfi & Galasso MVCHA 066	Mooka	Fabaceae	67	1	0.011	0.74	Bark	Lumpy skin disease	Decoction	Oral
^I^*Ziziphus mucronata* Willd. MVCHA 068	Sekgalo	Rhamnaceae	40	6	0.066	0.44	Roots	Dystocia, diarrhoea, arthralgia	Decoction	Oral
							Leaves	Wounds, foot rot	Ground	Topical
							Whole plant	Eye infection	Decoction	Oral

## Data Availability

Primary data collected during the survey are available upon request to the corresponding author of this manuscript.

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
