# Peer review of "Ethnoveterinary Practices and Ethnobotanical Knowledge on Plants Used against Cattle Diseases among Two Communities in South Africa"

_plants, 2022, doi:10.3390/plants11131784_

Round 1
Reviewer 1 Report
This article presented Ethnoveterinary practices and ethnobotanical knowledge on plants used against cattle diseases among two rural communities in South Africa. Before recommending this article for publication, there are some shortcomings that should be resolved.
General comments
Overall the experiment is well designed and presented in a good way but the English of the whole manuscript should be revised in terms of formatting and some typos. Check the species names in the whole MS must be italicized.
Abstract
In the abstract the authors should mention which potential therapeutic have been studied.
Remove repeated words in the whole MS and make the article consistent.
How literature etc was reviewed must be mentioned.
Conclude the findings and suggest future possibilities based on the findings of this study.
Introduction
Introduction is well written but some information could be further improved.
In Paragraph four add recent techniques, studies, and diversity of medicinal plants from recent studies i.e.
10.30848/PJB2022-3(19); 10.1016/j.jep.2021.114515
Also, I suggest to add significance of phytochemicals and traditional uses. The following studies can be cited.
Structure and length: The overall structure of the article is well organized and well balanced. The article is written with the minimum length necessary for all relevant information.
Logic: The article is written clearly and correctly. It is logically consistent.
Figures and tables: They are essential and clearly presented.
English: English used in the article should be improved. There are some typographic and grammatical errors.
Scientific quality rating: The article is novel and original. The article contains material that is new or adds significantly to knowledge already published.
Importance and impact: The presented results are of significant importance and impact to advancement in the relevant field of research.
References: Appropriate and adequate references to related works are covered sufficiently in the list. Incorporate some latest references as suggested above.
Author Response
"Please see the attachment."

Reviewer 2 Report
This is an interesting article about ethnoveterinary practices and ethnobotanical knowledge on plants used against cattle diseases among two rural communities in South Africa and it is worth being published in "Plants" after minor revisions. Data were collected from 90 participants and quantitatively analysed with three ethnobotanical indices (informant consensus factor, use-value and relative frequency of citation). About nearly 30% of 64 identified plants were documented as ethnoveterinary medicine for treating cattle ailments for the first time. This preserved knowledge is very valuable for further studies and the exploitation of new substances in the treatment of veterinary diseases. Documentation of the plant parts used, the method of preparation and route of administration of the plants used for treatment are highly appreciated. Material and Methods are described in understandable way and conclusions drawn.
Attached please find my minor corrections and questions in the text.

Author Response
"Please see the attachment."
